# Adherence to the 2017 French dietary guidelines and adult weight gain: A cohort study

**Dan Chaltiel**[1]*, **Chantal Julia**[1,2], **Moufidath Adjibade**[1], **Mathilde Touvier**[1],
**Serge Hercberg**[1,2], **Emmanuelle Kesse-Guyot**[1]

**1** Nutritional Epidemiology Research Team (EREN), Sorbonne Paris Cité Centre of Research in Epidemiology and Statistics (CRESS), Conservatoire National des Arts et Métiers, Paris 13 University, Bobigny, France, **2** Public Health Department, Avicenne Hospital, Assistance Publique–Hôpitaux de Paris, Bobigny, France

* d.chaltiel@eren.smbh.univ-paris13.fr

**Data Availability Statement:** Data of the study are protected under the protection of health data regulation set by the French National Commission for Information Technology and Liberties (Commission Nationale de l'Informatique et des

## Abstract

### Background

The French dietary guidelines were updated in 2017, and an adherence score to the new guidelines (Programme National Nutrition Santé Guidelines Score 2 [PNNS-GS2]) has been developed and validated recently. Since overweight and obesity are key public health issues and have been related to major chronic conditions, this prospective study aimed to measure the association between PNNS-GS2 and risk of overweight and obesity, and to compare these results with those for the modified Programme National Nutrition Santé Guidelines Score (mPNNS-GS1), reflecting adherence to 2001 guidelines.

### Methods and findings

Participants ($N$ = 54,089) were recruited among French adults ($\geq$18 years old, mean baseline age = 47.1 [SD 14.1] years, 78.3% women) in the NutriNet-Santé web-based cohort. Mean (SD) score was 1.7 (3.3) for PNNS-GS2 and 8.2 (1.6) for mPNNS-GS1. Selected participants were those included between 2009 and 2014 and followed up to September 2018 (median follow-up = 6 years). Collected data included at least three 24-hour dietary records over a 2-year period following inclusion, baseline sociodemographics, and anthropometric data over time. In Cox regression models, PNNS-GS2 was strongly and linearly associated with a lower risk of overweight and obesity (HR for quintile 5 versus quintile 1 [95% CI] = 0.48 [0.43–0.54], $p$ < 0.001, and 0.47 [0.40–0.55], $p$ < 0.001, for overweight and obesity, respectively). These results were much weaker for mPNNS-GS1 (HR for quintile 5 versus quintile 1 = 0.90 [0.80–0.99], $p$ = 0.03, and 0.98 [0.84–1.15], $p$ = 0.8, for overweight and obesity, respectively). In multilevel models, PNNS-GS2 was negatively associated with baseline BMI and BMI increase over time ($\beta$ for a 1-SD increase in score [95% CI] = −0.040 [−0.041; −0.038], $p$ < 0.001, and −0.00080 [−0.00094; −0.00066], $p$ < 0.001, respectively). In "direct comparison" models, PNNS-GS2 was associated with a lower risk of overweight and obesity, lower baseline BMI, and lower BMI increase over time than mPNNS-GS1. Study

Libertés, CNIL). The data are available upon request to the study's operational manager, Nathalie Pecollo (n.pecollo@eren.smbh.univ-paris13.fr), for review by the steering committee of the NutriNet-Santé study.

**Funding:** The NutriNet-Santé Study is supported by the French Ministry of Health (https://solidarites-sante.gouv.fr/ministere/organisation/directions/article/dgs-direction-generale-de-la-sante), the French Public Health Agency (http://www.santepubliquefrance.fr), the French National Institute for Health and Medical Research (INSERM, www.inserm.fr), the French National Institute for Agricultural Research (INRA, www.inra.fr), the National Conservatory for Arts and Crafts (CNAM, www.cnam.fr) and the Paris 13 University (www.univ-paris13.fr). Researchers were independent from funders and authors received no specific funding for this work. The funders had no role in study design, data collection and analysis, decision to publish, or preparation of the manuscript.

**Competing interests:** The authors have declared that no competing interests exist.

**Abbreviations:** AHEI-2010, Alternate Healthy Eating Index 2010; CU, consumption unit; FBDGs, food-based dietary guidelines; mPNNS-GS1, modified Programme National Nutrition Santé Guidelines Score; PNNS, Programme National Nutrition Santé; PNNS-GS1, Programme National Nutrition Santé Guidelines Score; PNNS-GS2, Programme National Nutrition Santé Guidelines Score 2; sPNNS-GS2, simplified Programme National Nutrition Santé Guidelines Score 2.

limitations include possible selection bias, reliance on participant self-report, use of arbitrary cutoffs in data analyses, and residual confounding, but robustness was tested in sensitivity analyses.

## Conclusions

Our findings suggest that adherence to the 2017 French dietary guidelines is associated with a lower risk of overweight and obesity. The magnitude of the association and the results of the direct comparison reinforced the validity of the updated recommendations.

## Trial registration

The NutriNet-Santé Study ClinicalTrials.gov (NCT03335644)

## Author summary

### Why was this study done?

- Obesity is a major condition, growing worldwide at a pandemic rate, and represents an important risk factor for main non-communicable diseases like cardiovascular diseases, type II diabetes, and some types of cancer.

- In public health, weight management is a critical lever to limit the occurrence of obesity in the population, and nutrition is an important part of it.

- French food-based dietary guidelines were updated in 2017, and a dietary score, PNNS-GS2, was developed to measure their level of adherence in the population.

- As part of evaluation of these guidelines, it was necessary to study the association between PNNS-GS2 and BMI change and the risk of developing overweight or obesity.

### What did the researchers do and find?

- We used statistical models to capture the effect of following the new dietary guidelines (reflected by a high PNNS-GS2) on weight.

- We used data from 54,089 participants of the French NutriNet-Santé cohort, with a median follow-up of 6 years.

- Having a high PNNS-GS2 was associated with lower weight gain and a lower risk of developing overweight or obesity.

- Another model allowed comparison of PNNS-GS2 to mPNNS-GS1, reflecting the former French dietary guidelines (2001), and PNNS-GS2 performed significantly better than its predecessor regarding the outcomes.

**What do these findings mean?**

- Following the 2017 French dietary guidelines can be expected to improve weight management in the French population.

- It is highly probable that this will also reduce the incidence of chronic diseases, as BMI is a strong risk factor.

- The next step will be to test the association of PNNS-GS2 with chronic diseases.

## Introduction

Worldwide, the prevalence of obesity nearly tripled between 1975 and 2019 and continues to grow at a pandemic rate [1]. However, in France, prevalence was 17% and 49% in 2015 for obesity and for overweight including obesity, respectively, and these numbers have been rather stable since 2006 [2]. The social cost of these 2 conditions in France was estimated to be 20 billion in 2012 [3].

In the past few years, the complex causes of obesity have been increasingly recognized, with the involvement of many dietary, behavioral, genetic, and environmental factors [4]. This is further illustrated by the classification International Classification of Diseases (ICD), which considered obesity as a unique condition in the 10th revision (code E66 in the ICD-10 of 2016 [5]) but as a wider category with several sub-items in the 11th revision (code 5B81 in the ICD-11 of 2018, which also includes overweight as code 5B80 [6]).

In addition, obesity has already been proven to increase the risk of numerous major chronic conditions like cardiovascular diseases, type II diabetes, Alzheimer disease, depression, and some types of cancer, and is associated with quality of life and all-cause mortality [1,4]. It is therefore widely accepted as an intermediate risk factor of major chronic diseases, and thus can be considered as a good marker of the overall health of the individual.

Indeed, overweight, and by extension obesity, is a multifactorial phenomenon, mostly caused by a chronic energy imbalance combining excessive caloric intake and insufficient energy expenditure [1]. The role of genetic susceptibility is well known, but the recent obesity pandemic might rather be attributable to environmental and lifestyle factors [7].

Thus, the rise in obesity incidence could be related to a decrease in physical activity level [8] combined with an increase in sedentary behaviors, but diet quality is also considered as a major determinant of body weight changes [9]. More recently, overweight and obesity were found to be negatively associated with organic food consumption [10], with a potential implication of pesticides [11].

In a recent meta-analysis based on 43 pooled prospective reports, the importance of diet quality was further reinforced, as risk of overweight, obesity, and/or weight gain was found to have a significant negative association with consumption of whole grains, fruits, nuts, legumes, and fish, and a significant positive association with consumption of refined grains, red meat, and sugary drinks [12].

Many studies have evaluated the relevance of national dietary recommendations by assessing the association between the level of adherence to specific recommendations and health outcomes [13–20]. Although some specificities exist (e.g., consideration of snacking in the Japanese guidelines) and scoring systems differ, most food-based dietary guidelines (FBDGs) include recommendations on main food groups, such as vegetables, fruits, grains, meat, and

alcohol. To assess the level of adherence to such dietary guidelines, these studies [13–20] have developed predefined dietary scores. The magnitude of the association between health and dietary score varies noticeably depending on the underlying guidelines and on the score construction methodology.

In March 2017, French FBDGs were revised [21] in preparation for the fourth iteration of the French National Nutrition and Health Program (Programme National Nutrition Santé [PNNS]). We recently developed and validated a dietary index, the PNNS Guidelines Score 2 (PNNS-GS2), estimating the adherence to these new recommendations [22]. This index aimed at updating the PNNS Guidelines Score, which was based on the 2001 FBDGs and will be referred to here as PNNS-GS1 for clarity [13]. It should be noted that PNNS-GS1, unlike PNNS-GS2, included "physical activity" as a component.

Therefore, the present study aimed to assess, in a large French cohort, the prospective associations between PNNS-GS2 and the incidence of overweight and obesity compared to those observed with the mPNNS-GS1 (a modified version of PNNS-GS1 without physical activity) to assess the validity of the updated FBDGs.

## Methods

### Study population

The data were extracted from the NutriNet-Santé cohort, which is a large, ongoing, web-based observational cohort launched in France in 2009. It aims to investigate the relationship between nutrition and health, along with determinants of dietary behavior and nutritional status, and the full design and methodology have been described elsewhere [23]. Participants were recruited through vast multimedia campaigns among the adult (≥18 years old) population with access to the internet. All questionnaires were pilot-tested and completed online using a dedicated website (https://www.etude-nutrinet-sante.fr). The NutriNet-Santé study is conducted in accordance with the Declaration of Helsinki and was approved by the ethics committee of the French Institute for Health and Medical Research (IRB Inserm no. 0000388FWA00005831) and by the National Commission on Informatics and Liberty (CNIL no. 908450 and no. 909216). Electronic informed consent was obtained from all participants. The NutriNet-Santé study is registered in ClinicalTrials.gov (NCT03335644). All questionnaires can be accessed at https://info.etude-nutrinet-sante.fr/en/node/11. The data of the study are protected under the protection of health data regulation set by the CNIL; however, they are available upon request for review by the steering committee of the NutriNet-Santé study. Analyses were hypothesis-oriented based on the relationship between dietary scores and weight gain or prospective occurrence of overweight or obesity. A single non-prespecified analysis was performed to investigate an unexpected result. All methods have been described in line with the Strengthening the Reporting of Observational Studies in Epidemiology (STROBE) Statement (see S1 STROBE Checklist).

### Dietary data

Participants in the NutriNet-Santé cohort provide, at baseline then twice a year, 3 non-consecutive 24-hour dietary records assigned over a 2-week period. The 3 recording days are randomly assigned to 2 weekdays and 1 weekend day to account for intra-individual variability in intake. All food and drink consumption throughout the entire day (midnight to midnight) was recorded by participants via a dedicated online platform providing a food browser (grouped by category) and a search engine that allows searching for any food by name. Participants declared consumed amounts as absolute units when known (in grams or milliliters), using

common household measures, or using generic food portion size from previously validated pictures [24].

Consumptions were weighted according to weekday versus weekend day, and daily energy and nutrient intakes were computed using validated and constantly updated composition tables including more than 3,000 food items [25]. Under-reporting was identified using the Black method [26]. This dietary recording protocol has been tested and validated against an interview by a trained dietitian and against blood and urinary biomarkers [27–29].

Within 2 months after participant inclusion, frequency of organic food consumption was assessed for fruits, vegetables, bread, and starchy foods (rice, pasta, and legumes) using a previously described questionnaire [30]. Frequencies were assessed using 3 modalities of consumption: (1) most of the time, (2) occasionally, and (3) never (with detailed reasons that are not considered here). Concerning starchy foods, the frequency of organic food consumption was considered twice, once for rice and pasta and once for legumes, but each item was considered null if it was not reported as consumed in the 24-hour dietary records.

### Outcome data

Height and weight were self-reported at enrollment and at least yearly thereafter using a web-based anthropometric questionnaire [31]. Participants were asked to assess these data during a medical or occupational health examination by a physician, or by self-measurement using standardized procedures (on flat surface, lightly dressed, and without shoes). Although self-reported, these data have been validated against clinical measures, with intraclass correlation coefficients of 0.94 and 0.99 for height and weight, respectively, and a correct BMI classification in 93% of cases [32]. BMI (kg/m$^2$) was computed by dividing weight by height squared. Overweight was defined as having a BMI $\geq$ 25 kg/m$^2$, and obesity as having a BMI $\geq$ 30 kg/m$^2$, following the World Health Organization reference values [9].

### Other data

Participants filled in their sociodemographic and lifestyle characteristics (age, sex, education, occupation, income, cohabitating status, physical activity, and smoking habits) using a dedicated self-administered web-based questionnaire [23]. Physical activity was assessed by the International Physical Activity Questionnaire (IPAQ) [33]. Income was estimated monthly per consumption unit (CU) according to a weighting system where 1 CU is attributed for the first adult in the household, 0.5 CU for other persons aged 14 years or older, and 0.3 CU for children under 14 years [34]. All questionnaires are available at https://info.etude-nutrinet-sante.fr/en/node/11.

### Sample selection

Participants were drawn from the NutriNet-Santé cohort (*N* = 138,014). Data used in the present paper were based on participants included between 7 March 2009 and 12 December 2014 and followed up until 30 September 201 at the most. A detailed flowchart is presented in Fig 1. Exclusions led to a working sample of 54,089 participants, and analyses on overweight and obesity were performed on 32,954 and 44,026 participants without overweight and obesity at baseline, respectively. To assess potential selection bias, the working sample was compared to the whole NutriNet-Santé cohort regarding sociodemographics. We also conducted a sensitivity analysis on a subpopulation ("sensitivity subpopulation") excluding any participant who had a cancer diagnosis during follow-up (*N* = 1,981), bariatric surgery (*N* = 90) or an eating disorder (*N* = 2,153), or extreme BMI (as per the first percentile, *N* = 790).

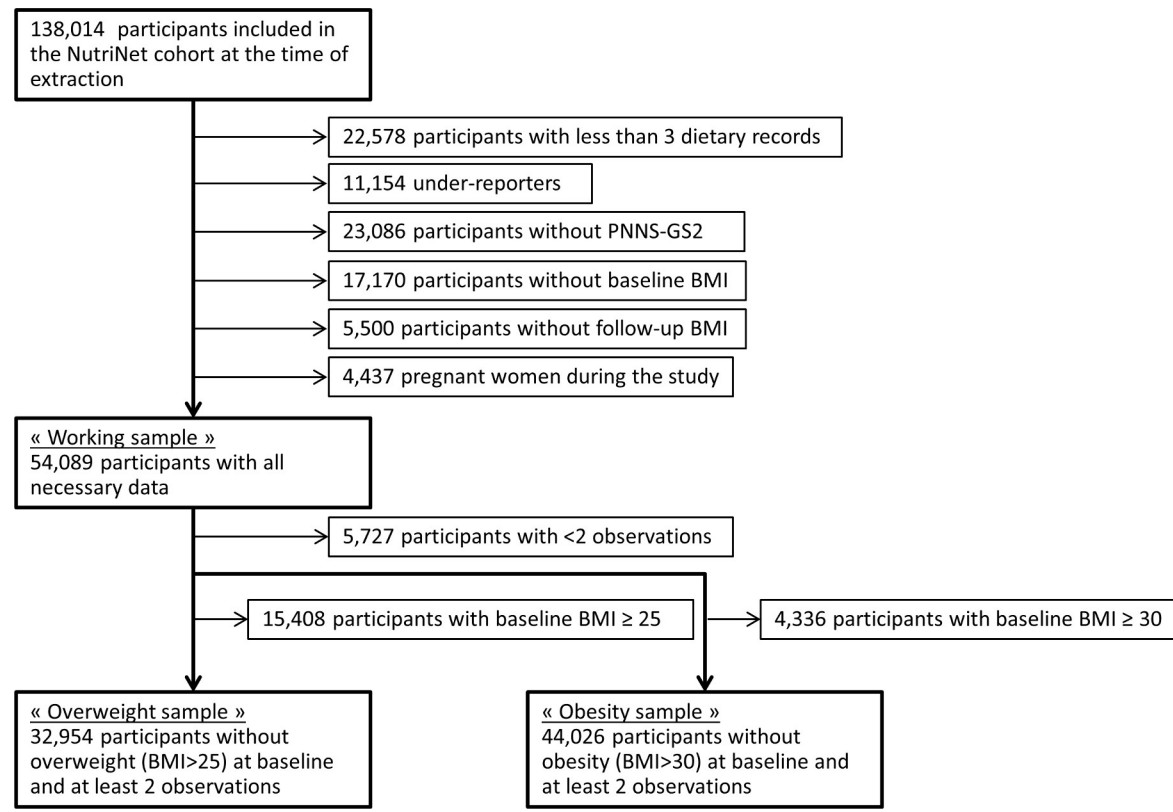

**Fig 1. Flowchart of participants included in the present analysis of the NutriNet-Santé cohort.** PNNS-GS2, Programme National Nutrition Santé Guidelines Score 2.

## Computation of dietary scores

PNNS-GS2 is a dietary index designed to reflect adherence to the 2017 French FBDGs [21,35], whereas mPNNS-GS1 is a modified version of the index based on the 2001 FBDGs minus physical activity. PNNS-GS2 includes 13 components (6 of adequacy and 7 of moderation), while mPNNS-GS1 includes 12 components (7 of adequacy and 5 of moderation). Their components, scorings, and weights are presented in Table 1. sPNNS-GS2 is a simplified version of PNNS-GS2, constructed in the same way but considering only the principal recommendations. The validation process of PNNS-GS2 and sPNNS-GS2 has been thoroughly described elsewhere [13,22]. Dietary scores were computed from average consumption estimated using all 24-hour dietary records completed in the first 2 years after inclusion, and thus considered as the usual diet.

For comparison with international data, we also computed the Alternate Healthy Eating Index 2010 (AHEI-2010, minus trans fatty acids, which were unavailable in our cohort), which is based on a comprehensive review of the relevant literature in order to best predict the risk of chronic diseases [14]. AHEI-2010, ranging from 0 to 100, is the sum of 10 components, ranging from 0 to 10 points each, which are considered either "desirable," thus contributing positively to the score (vegetables, fruits, whole grains, nuts and legumes, long-chain ω-3 fatty acids, polyunsaturated fatty acids, and moderate alcohol consumption) or "undesirable," thus contributing negatively to the score (sodium, sugar-sweetened drinks and fruit juice, and red and processed meat).

For comparability, all scores were standardized by dividing by their standard deviation.

**Table 1. PNNS-GS1 and PNNS-GS2: Comparison of components and scoring.** [a]

| Dietary component[b] | PNNS-GS1 | | | PNNS-GS2 | | |
|---|---|---|---|---|---|---|
| | Recommendation | Criteria[c] | Score | Recommendation[d] | Criteria[c] | Score |
| Fruits and vegetables (weight = 3) | At least 5 servings/day, with 1 max as juice and 1 max as dried | [0–3.5) | 0 | **At least 5 servings/day, with 1 max as juice and 1 max as dried** | [0–3.5) | 0 |
| | | [3.5–5) | 0.5 | | [3.5–5) | 0.5 |
| | | [5–7.5) | 1 | | [5–7.5) | 1 |
| | | ≥7.5 | 2 | | ≥7.5 | 2 |
| | | — | | Prefer organic fruits | Most of the time | 0.5 |
| | | | | | Occasionally | 0.25 |
| | | | | | Never | 0 |
| | | — | | Prefer organic vegetables | Most of the time | 0.5 |
| | | | | | Occasionally | 0.25 |
| | | | | | Never | 0 |
| Nuts (weight = 1) | | — | | **A handful/day** | 0 | 0 |
| | | | | | (0–0.5) | 0.5 |
| | | | | | [0.5–1.5) | 1 |
| | | | | | ≥1.5 | 0 |
| Legumes (weight = 1) | | — | | **At least 2 servings/week** | 0/week | 0 |
| | | | | | (0–2)/week | 0.5 |
| | | | | | ≥2/week | 1 |
| | | — | | Prefer organic legumes | Most of time | 0.5 |
| | | | | | Occasionally | 0.25 |
| | | | | | Never | 0 |
| Bread, cereals, potatoes, and legumes | At each meal according to appetite | [0–1) | 0 | | — | |
| | | [1–3) | 0.5 | | | |
| | | [3–6) | 1 | | | |
| | | ≥6 | 0.5 | | | |
| Whole-grain food (weight = 2) | Preferentially choose whole grains and whole-grain breads | [0–1/3) | 0 | **Every day** | 0 | 0 |
| | | [1/3–2/3) | 0.5 | | (0–1) | 0.5 |
| | | ≥2/3 | 1 | | [1–2) | 1 |
| | | | | | ≥2 | 1.5 |
| | | — | | Prefer organic bread | Most of the time | 0.5 |
| | | | | | Occasionally | 0.25 |
| | | | | | Never | 0 |
| | | — | | Prefer organic grains | Most of the time | 0.5 |
| | | | | | Occasionally | 0.25 |
| | | | | | Never | 0 |
| Milk and dairy products (weight = 1) | 3 per day (≥55 years: 3 to 4 per day) | [0–1) | 0 | **2 servings/day** | [0–0.5) | 0 |
| | | [1–2.5) | 0.5 | | [0.5–1.5) | 0.5 |
| | | [2.5–3.5] (≥55 years old: [2.5–4.5]) | 1 | | [1.5–2.5) | 1 |
| | | >3.5 (≥55 years old: >4.5) | 0 | | ≥2.5 | 0 |
| Red meat (weight = 2) | | — | | **Limit consumption** | [0–500) g/week | 0 |
| | | | | | [500–750) g/week | −1 |
| | | | | | ≥750 g/week | −2 |

*(Continued)*

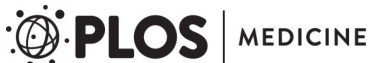

**Table 1.** (*Continued*)

| Dietary component[b] | PNNS-GS1 | | | PNNS-GS2 | | |
|---|---|---|---|---|---|---|
| | Recommendation | Criteria[c] | Score | Recommendation[d] | Criteria[c] | Score |
| Processed meat (weight = 3) | | — | | **Limit consumption** | [0–150) g/week | 0 |
| | | | | | [150–300) g/week | −1 |
| | | | | | ≥300 g/week | −2 |
| | | — | | Prefer white ham over other processed meat[e] | Ratio <50% | 0 |
| | | | | | Ratio ≥50% | 0.5 |
| Meat, poultry, seafood, and eggs | 1 to 2 per day | 0 | 0 | | — | |
| | | (0–1) | 0.5 | | | |
| | | [1–2] | 1 | | | |
| | | >2 | 0 | | | |
| Fish and seafood (weight = 2) | At least twice a week | <2 servings per week | 0 | **2 servings/week** | [0–1.5) servings/week | 0 |
| | | ≥2 servings per week | 1 | | [1.5–2.5) servings/week | 1 |
| | | | | | [2.5–3.5) servings/week | 0.5 |
| | | | | | ≥3.5 servings/week | 0 |
| | | — | | Fatty fish 1 serving/week | [0–0.5) servings/week | 0 |
| | | | | | [0.5–1.5) servings/week | 1 |
| | | | | | ≥1.5 servings/week | 0 |
| Added fat (weight = 2) | Limit consumption; prefer vegetal fat over animal fat | >16% of EIWA | 0 | **Limit consumption** | >16% of EIWA | 0 |
| | | ≤16% of EIWA | 1 | | ≤16% of EIWA | 1.5 |
| | | Ratio > 50% | 0 | Prefer vegetal fat over animal fat | Ratio > 50% | 0 |
| | | Ratio ≤ 50% | 1 | | Ratio ≤ 50% | 1 |
| | | | | Prefer ALA-rich and olive oils over other oils | Ratio < 50% | 0 |
| | | | | | Ratio ≥ 50% | 1 |
| Sugary foods (weight = 3) | Limit consumption | <10% of EIWA | 1 | **Limit consumption** | <10% of EIWA | 0 |
| | | [10%–15%) of EIWA | 0 | | [10%–15%) of EIWA | −1 |
| | | ≥15% of EIWA | −0.5 | | ≥15% of EIWA | −2 |
| Sweet-tasting beverages[f] (weight = 3) | Drink water as desired; limit sweetened beverages: no more than 1 glass/day | <1 l water and >250 ml soda/day | 0 | **Limit consumption** | 0 ml/day | 0 |
| | | ≥1 l water and >250 ml soda/day | 0.50 | | (0–250) ml/day | −0.5 |
| | | <1 l water and ≤ 250 ml soda/day | 0.75 | | [250–750) ml/day | −1 |
| | | ≥1 l water and ≤ 250 ml soda/day | 1 | | ≥750 ml/day | −2 |
| Alcoholic beverages (weight = 3) | ≤2 glasses of wine/day for women and ≤3 glasses/day for men | Ethanol >20 (♀) or >30 (♂) g/day | 0 | **Limit consumption** | 0 g/day | 0.5 |
| | | Ethanol ≤20 (♀) or ≤30 (♂) g/day | 0.8 | | (0–100] g/day | 0 |
| | | Abstainers and irregular consumers (<once a week) | 1 | | (100–200] g/day | −1 |
| | | | | | >200 g/day | −2 |

(*Continued*)

**Table 1.** (Continued)

| Dietary component[b] | PNNS-GS1 | | | PNNS-GS2 | | |
|---|---|---|---|---|---|---|
| | Recommendation | Criteria[c] | Score | Recommendation[d] | Criteria[c] | Score |
| Salt (weight = 3) | Limit consumption | ≤6 g/day | 1.5 | **Limit consumption** | ≤6 g/day | 1 |
| | | (6–8] g/day | 1 | | (6–8] g/day | 0 |
| | | (8–10] g/day | 0.5 | | (8–10] g/day | −0.5 |
| | | (10–12] g/day | 0 | | (10–12] g/day | −1 |
| | | >12 g/day | −0.5 | | >12 g/day | −2 |
| Physical activity | At least the equivalent of 30 min of brisk walking per day | [0–30) min/day | 0 | — | | |
| | | [30–60) min/day | 1 | | | |
| | | ≥60 min/day | 1.5 | | | |

[a]Exact similarities between PNNS-GS1 and PNNS-GS2 are shaded gray.

[b]Weights are considered only in PNNS-GS2.

[c]Servings per day unless otherwise is stated; brackets indicate inclusive range limits; parentheses indicate exclusive range limits.

[d]PNNS-GS2 principal components are in bold.

[e]Conditional: The 0.5 bonus point only occurs if total processed meat consumption is more than 150 g/week.

[f]Sweetened beverages are specifically sweet beverages and, for PNNS-GS2 only, artificially sweetened beverages and fruit juices.

ALA, α-linolenic acid; EIWA, energy intake without alcohol; max, maximum; PNNS-GS1, Programme National Nutrition Santé Guidelines Score; PNNS-GS2, Programme National Nutrition Santé Guidelines Score 2.

## Statistical analysis

For descriptive purposes, quintiles of PNNS-GS2 and mPNNS-GS1 were calculated for men and women separately. Associations of baseline sociodemographics across quintiles were tested with a chi-squared test for unordered factors with more than 2 levels, with a Cochran–Armitage test for trend for unordered factors with 2 levels exactly, with a Spearman correlation test for ordered factors, and with a linear contrast test for numeric variables. The baseline time was defined as the median time between the first and the last dietary record, which represents our exposure window.

Next, we assessed the association of PNNS-GS2, sPNNS-GS2, and mPNNS-GS1 with changes in BMI over time and the risk of overweight and obesity.

First, we estimated the association between the dietary scores and changes in BMI over time with linear multilevel mixed models in the working sample ($N$ = 54,089). Fixed effects of dietary scores (in quintiles and continuous), time, and their interaction were entered in the model, with a random effect for participant and time. BMI and energy intake without alcohol were considered as their natural logarithm to improve model fit, and models' residuals were graphically tested. Time was calculated as the difference between the date of an anthropometric questionnaire and the baseline date. Spatial power law SP(POW)(time) from SAS MIXED procedure was used because the correlation changed as a function of time, since the time interval between each questionnaire was not constant. Different variance–covariance matrix structures were tested, and the best one (unstructured) was selected based on AIC.

Models were first adjusted for sex, age, energy intake without alcohol, and number of completed 24-hour dietary records, with second- and third-order interactions between score, time, and sex (model "m0"), and then further adjusted for height, month of inclusion (in order to take into account potential seasonal effects), physical activity (in metabolic equivalents [36]), occupation (8 categories as per the French National Institute of Statistics and Economic Studies classification [37]), smoking status (non-smoker, former smoker, or smoker), educational

level (primary, secondary, or university), monthly income (≤1,800 /CU, 1,800–2,700 /CU, or >2,700 /CU), and cohabiting status (model "m1").

Next, we estimated the association between dietary scores (quintiles and continuous after standardization) and incidence of overweight and obesity using a multivariable Cox proportional hazard model. As the inclusion time did not represent a specific event, common to all participants, that could be considered as "zero" time, we decided to use age as the timescale [38,39]. People with overweight or obesity at baseline were removed from the prospective analyses. Events were defined as the first occurrence of BMI $\geq 25$ kg/m$^2$ for overweight and of BMI $\geq 30$ kg/m$^2$ for obesity. Participants contributed follow-up time from their entry in the study until the midpoint between the questionnaire where the first event was reported and the previous one, or until their last questionnaire if no event was reported, so that each person contributed only 1 endpoint to the analysis. The data were thus left-truncated and right-censored. Linear trend across dietary score quintiles was tested by assigning the median value in each quintile, which was then entered as a continuous variable in the model. Models were adjusted as in the multilevel models, except that age is already handled in the timescale, and that interaction with sex was not significant and was therefore removed.

Each continuous-adjustment covariable was corrected for log-linearity using restricted cubic splines with 3 nodes [40] using the *rms* package for R [41]. The proportional hazard assumption was tested graphically using Schoenfeld residuals and the Grambsch–Therneau test [42]. All analyses were performed in men and women altogether as no interaction with sex was detected.

As a sensitivity analysis, we replicated these Cox models within the sensitivity subpopulation (excluding any case of cancer, bariatric surgery or eating disorder, or extreme BMI), with the same methodology.

For both sets of analyses (multilevel and Cox models), since models were not nested, direct comparison of the predictive value of PNNS-GS2, sPNNS-GS2, and mPNNS-GS1 for BMI evolution or overweight and obesity risk was not possible. Therefore, we added standardized (by dividing by their standard deviation) scores in pairs in fully adjusted models and used a Wald test to assess if the respective score-related coefficients were significantly different from each other, similarly to Chiuve et al. [14]. This technique is referred to as the "one-model comparison of scores" hereafter. Resulting hazard ratio magnitudes are not to be interpreted directly, but the direction of associations should provide information in the qualitative (superiority/inferiority) comparison of these scores.

Since less than 1% of participants in the working sample had missing data, all analyses have been performed on complete cases.

All statistical analyses were conducted using R (version 3.4.2) and SAS (version 9.4) with a significance level of 5% for 2-sided tests.

## Results

The working sample was composed of 41,164 (76.1%) women and 12,925 (23.9%) men, who completed in average 7.7 (SD 2.3) 24-hour dietary records per person. During the follow-up, the participants provided on average 11.4 (SD 4.7) weight values. Participants were on average 47.1 (SD 14.1) years old at inclusion.

In the working sample, mean (SD) score was 1.7 (3.3) for PNNS-GS2 and 8.2 (1.6) for mPNNS-GS1. Median follow-up was 6.0 years for both the overweight and the obesity study, and median participation time was 5.5 years for the BMI variation analysis.

Associations of PNNS-GS2 with baseline covariables and food group consumptions are presented respectively in Tables 2 and 3. Covariable associations with mPNNS-GS1 are presented

**Table 2. Baseline characteristics of the participants by quintile of PNNS-GS2—NutriNet-Santé study, N = 54,089[a].**

| Characteristic | Total population | Dietary score quintile | | | | |
|---|---|---|---|---|---|---|
| | | Q1 | Q2 | Q3 | Q4 | Q5 |
| PNNS-GS2 | 1.7 (3.3) | −3.0 (1.9) | −0.2 (1.3) | 1.6 (1.2) | 3.3 (1.0) | 6.0 (1.5) |
| sPNNS-GS2 | 1.9 (3.5) | −2.9 (2.3) | 0.0 (1.7) | 1.8 (1.6) | 3.5 (1.5) | 6.0 (1.8) |
| mPNNS-GS1 | 8.1 (1.6) | 6.6 (1.4) | 7.5 (1.4) | 8.1 (1.3) | 8.7 (1.3) | 9.4 (1.3) |
| Age at inclusion (years)[b] | 47.1 (14.1) | 43.3 (14.1) | 45.6 (14.3) | 47.1 (14.1) | 48.5 (13.9) | 50.1 (13.4) |
| Number of weight measurements[b] | 11.4 (4.7) | 10.9 (4.8) | 11.2 (4.7) | 11.5 (4.7) | 11.5 (4.7) | 11.7 (4.7) |
| Height (cm)[b] | 166.5 (8.2) | 167.7 (8.3) | 167.0 (8.2) | 166.4 (8.1) | 166.0 (8.1) | 165.6 (8.1) |
| Body mass index (kg/m$^2$)[b] | 23.9 (4.4) | 24.8 (5.2) | 24.2 (4.6) | 23.9 (4.4) | 23.6 (4.1) | 23.0 (3.8) |
| Energy intake without alcohol (kcal/day)[b] | 1,821.7 (441.8) | 2,060.6 (462.4) | 1,908.9 (442.8) | 1,807.3 (404.1) | 1,730.1 (404.2) | 1,653.9 (388.0) |
| Ethanol consumption (g/day)[b] | 8.3 (11.5) | 15.2 (16.8) | 9.8 (12.2) | 7.7 (10.0) | 5.9 (7.5) | 4.1 (5.9) |
| Physical activity (METs)[b] | 2,866 (2,773) | 2,660 (2,743) | 2,713 (2,704) | 2,781 (2,690) | 2,932 (2,784) | 3,182 (2,892) |
| Sex[c] | | | | | | |
| Female | 76.1% | 75.0% | 75.6% | 76.1% | 76.4% | 77.1% |
| Male | 23.9% | 25.0% | 24.4% | 23.9% | 23.6% | 22.9% |
| Education[d] | | | | | | |
| Primary | 1.0% | 1.1% | 1.2% | 0.8% | 1.1% | 0.8% |
| Secondary | 35.5% | 38.3% | 36.0% | 36.0% | 34.2% | 33.8% |
| University | 63.5% | 60.5% | 62.9% | 63.2% | 64.7% | 65.4% |
| Occupational category[e] | | | | | | |
| Farmer/self-employed | 1.9% | 2.4% | 2.0% | 1.8% | 1.8% | 1.9% |
| Managerial staff | 23.0% | 21.3% | 22.2% | 22.7% | 24.0% | 24.2% |
| Employee | 16.1% | 19.3% | 17.5% | 16.7% | 14.4% | 13.2% |
| Student | 5.9% | 7.7% | 6.6% | 6.1% | 5.0% | 4.3% |
| Manual worker | 1.0% | 1.8% | 1.3% | 0.9% | 0.7% | 0.6% |
| Intermediate profession | 17.1% | 17.9% | 17.3% | 17.6% | 17.0% | 16.0% |
| Retired | 23.8% | 17.4% | 21.8% | 23.6% | 26.1% | 28.6% |
| Unemployed | 11.2% | 12.2% | 11.4% | 10.6% | 10.8% | 11.3% |
| Income[d] | | | | | | |
| ≤1,800 /CU | 43.3% | 50.5% | 46.3% | 43.2% | 40.2% | 38.1% |
| 1,800–2,700 /CU | 27.0% | 24.8% | 26.5% | 27.0% | 27.8% | 28.4% |
| >2,700 /CU | 29.7% | 24.7% | 27.2% | 29.9% | 32.0% | 33.5% |
| Smoking[d] | | | | | | |
| Non-smoker | 49.9% | 43.4% | 48.6% | 50.6% | 51.6% | 54.0% |
| Former smoker | 36.5% | 35.2% | 35.7% | 36.4% | 37.1% | 37.7% |
| Smoker | 13.6% | 21.4% | 15.8% | 13.0% | 11.3% | 8.3% |
| Cohabiting status[c] | | | | | | |
| Living alone | 27.6% | 26.2% | 26.2% | 26.7% | 28.1% | 30.3% |
| Cohabiting | 72.4% | 73.8% | 73.8% | 73.3% | 71.9% | 69.7% |

[a]Values are percentages or mean (standard deviation), as appropriate. All $p$-values from specified tests were <0.001.

[b]Linear contrast trend test.

[c]Cochran–Armitage trend test.

[d]Spearman correlation test for ordinal variables.

[e]Pearson chi-squared association test.

CU, consumption unit; METs, metabolic equivalents; mPNNS-GS1, modified Programme National Nutrition Santé Guidelines Score; PNNS-GS2, Programme National Nutrition Santé Guidelines Score 2; sPNNS-GS2, simplified Programme National Nutrition Santé Guidelines Score 2.

**Table 3. Food group consumption by quintile of PNNS-GS2, NutriNet-Santé study, *N* = 54,089[a].**

| Characteristic | Total population | Dietary score quintile | | | | | Correlation coefficient (95% CI)[b] |
|---|---|---|---|---|---|---|---|
| | | Q1 | Q2 | Q3 | Q4 | Q5 | |
| Fruits (g/day) | 212.3 (142.1) | 128.4 (108.8) | 175.0 (119.6) | 203.3 (122.4) | 235.2 (131.5) | 298.9 (157.3) | **0.406 (0.399; 0.413)** |
| Vegetables (g/day) | 227.5 (107.0) | 178.1 (92.0) | 204.7 (95.8) | 221.3 (94.9) | 242.2 (105.2) | 279.2 (114.7) | **0.320 (0.312; 0.327)** |
| Fruit juices (ml/day) | 50.8 (68.2) | 51.6 (75.9) | 53.3 (71.4) | 52.5 (66.9) | 50.7 (65.1) | 46.5 (62.2) | −0.027 (−0.036; −0.019) |
| Vegetable juices (ml/day) | 1.8 (13.3) | 1.2 (11.6) | 1.3 (10.5) | 1.4 (10.8) | 1.8 (13.9) | 3.0 (17.7) | 0.048 (0.039; 0.056) |
| Legumes (g/day) | 12.0 (19.9) | 7.6 (16.1) | 9.2 (16.4) | 10.5 (16.5) | 12.2 (18.6) | 19.1 (26.7) | **0.199 (0.190; 0.207)** |
| Potatoes and other tubers (g/day) | 46.4 (37.0) | 48.2 (40.4) | 45.6 (37.0) | 45.9 (35.5) | 46.4 (35.9) | 46.1 (36.6) | −0.013 (−0.022; −0.005) |
| Whole-grain cereals (g/day) | 36.2 (43.2) | 17.3 (29.7) | 26.4 (35.9) | 32.5 (38.1) | 40.7 (42.4) | 59.2 (51.8) | **0.329 (0.322; 0.337)** |
| Refined cereals (g/day) | 140.6 (65.9) | 146.2 (67.8) | 147.1 (67.7) | 144.9 (64.7) | 141.0 (64.1) | 126.2 (63.4) | −0.107 (−0.116; −0.099) |
| Breakfast cereals (g/day) | 6.3 (13.9) | 4.0 (11.7) | 5.2 (12.6) | 5.9 (12.8) | 7.2 (14.8) | 8.8 (15.8) | 0.121 (0.113; 0.129) |
| Milk and dairy products (ml and g/day) | 234.3 (142.3) | 232.4 (148.4) | 239.5 (142.7) | 245.3 (142.9) | 242.1 (140.6) | 213.4 (135.6) | −0.044 (−0.052; −0.036) |
| Eggs (g/day) | 14.1 (15.6) | 13.2 (16.3) | 13.8 (15.7) | 14.1 (15.5) | 14.4 (15.3) | 15.0 (15.5) | 0.036 (0.028; 0.044) |
| Fish and seafood (g/day) | 40.0 (33.7) | 34.3 (34.1) | 38.3 (33.6) | 41.1 (33.7) | 42.5 (33.7) | 42.6 (32.6) | 0.077 (0.069; 0.086) |
| Meat (g/day) | 69.5 (41.4) | 85.3 (47.3) | 74.8 (41.3) | 70.2 (38.0) | 65.8 (37.7) | 55.0 (37.2) | **−0.249 (−0.257; −0.241)** |
| Processed meat/fish (g/day) | 33.5 (25.8) | 54.4 (30.2) | 41.2 (26.0) | 32.8 (22.0) | 25.8 (18.7) | 17.8 (15.1) | **−0.475 (−0.482; −0.469)** |
| Fatty, sweet, or salty food (g/day) | 100.9 (47.6) | 103.4 (55.3) | 105.1 (50.6) | 104.1 (46.9) | 100.0 (43.7) | 93.2 (41.2) | −0.073 (−0.081; −0.064) |
| Olive oil (g/day) | 4.6 (5.6) | 3.0 (5.0) | 3.8 (5.2) | 4.4 (5.3) | 5.1 (5.5) | 6.4 (6.3) | **0.204 (0.196; 0.212)** |
| Other oils (g/day) | 4.5 (4.9) | 4.9 (5.4) | 4.4 (4.9) | 4.4 (4.6) | 4.2 (4.6) | 4.5 (5.1) | −0.019 (−0.027; −0.010) |
| Other added fats (g/day) | 63.3 (46.5) | 57.9 (45.2) | 61.9 (45.1) | 64.7 (46.9) | 66.0 (46.9) | 65.2 (47.9) | 0.049 (0.041; 0.058) |
| Unsweetened drinks (ml/day) | 1,125.6 (501.5) | 1,040.8 (511.5) | 1,095.5 (503.8) | 1,120.8 (488.6) | 1,150.3 (490.9) | 1,201.0 (500.7) | **−0.326 (−0.333; −0.318)** |
| Sweetened beverages (ml/day) | 35.9 (76.0) | 66.2 (130.7) | 39.6 (74.7) | 31.3 (57.7) | 26.6 (48.7) | 21.3 (36.9) | 0.104 (0.096; 0.113) |
| Alcoholic beverages (ml/day) | 99.9 (133.4) | 179.9 (193.7) | 116.7 (141.5) | 93.5 (114.2) | 72.5 (88.8) | 53.2 (74.7) | **−0.200 (−0.209; −0.192)** |

[a]Values are given per day, as mean (standard deviation), adjusted for energy intake and sex using the residual method.

[b]Pearson's correlation coefficients with confidence intervals. All correlations were significantly different from 0 ($p < 0.001$). Absolute values > 0.2 are in bold.

in S1 Table. Higher adherence with French FBDGs (both 2001 and 2017) was positively associated with age, education, income, cohabiting status, and physical activity, and negatively associated with baseline BMI, energy intake without alcohol, alcohol consumption, and smoking. Unexpectedly, height was associated with PNNS-GS2. This association did not persist in an ancillary analysis adjusting a linear model of height as a function of PNNS-GS2 on age, sex, and energy intake without alcohol (unadjusted coefficient for PNNS-GS2: −0.66, $p < 0.001$; adjusted coefficient: −0.01, $p = 0.09$). By design, PNNS-GS2 was positively associated with higher consumption of fruits, vegetables, legumes, and whole-grain cereals and higher frequency of organic food consumption, and negatively associated with higher consumption of red and processed meat, refined cereals, and sweetened and alcoholic drinks.

Comparison of our selected working sample with the whole NutriNet-Santé cohort is presented in S2 Table. The selected population was significantly older, more physically active, more often male, less often smokers, and more often cohabiting, with a lower BMI at baseline, a better education, and a higher income.

The results of the longitudinal association between dietary score and the evolution of BMI using multilevel model regressions are presented in Table 4. After adjustment for confounding variables in the model m0, both higher adherence to 2017 FBDGs (measured by PNNS-GS2) and higher adherence to 2001 FBDGs (measured by mPNNS-GS1) were significantly associated with a lower BMI at baseline and with a lower increase of BMI over time. In the model

**Table 4. Longitudinal evolution of log(BMI) as a function of PNNS-GS2 and of mPNNS-GS1—NutriNet-Santé study.**

| Model | PNNS-GS2 | | mPNNS-GS1 | |
|---|---|---|---|---|
| | β (95% CI)[a] | p-Value[b] | β (95% CI)[a] | p-Value[b] |
| **m0[c]** | | | | |
| Score (1 SD) | −0.040 (−0.041; −0.038) | <0.001 | −0.0027 (−0.0044; −0.0010) | 0.002 |
| Time (years) | 0.0027 (0.0025; 0.0029) | <0.001 | 0.0049 (0.0043; 0.0056) | <0.001 |
| Score × time | −0.0008 (−0.00094; −0.00066) | <0.001 | −0.00055 (−0.00068; −0.00042) | <0.001 |
| **m1[d]** | | | | |
| Score (1 SD) | −0.037 (−0.038; −0.035) | <0.001 | 0.0018 (0.000090; 0.0035) | 0.043 |
| Time (years) | 0.0027 (0.0025; 0.0028) | <0.001 | 0.0049 (0.0043; 0.0056) | <0.001 |
| Score × time | −0.00075 (−0.00090; −0.00060) | <0.001 | −0.00055 (−0.00068; −0.00042) | <0.001 |

[a]Coefficients β were computed using a linear multilevel mixed model expressing the relationship between log(BMI) and PNNS-GS2 or mPNNS-GS1 score (expressed as number of standard deviations) and time (in years). Logarithm of BMI was used to increase normality and model's residual fitness. The coefficient for score represents the association of FBDGs with baseline BMI, the coefficient for time represents the mean evolution of BMI over time, and the coefficient for the interaction term represents the association of FBDGs with evolution of BMI over time. As BMI was log-transformed, interpretation of the interaction term is less intuitive since additive effects on log(BMI) become multiplicative on BMI, e.g., for an increase of 2 SD of score and 5 years, BMI is multiplied by $\exp(2\beta_{score} + 5\beta_{time} + 5 \times 2 \times \beta_{score \times time})$.

[b]p-Values were computed using a Wald test for coefficient nullity.

[c]m0 is the base model, adjusted for sex, age, energy intake without alcohol, and number of completed 24-hour dietary records, with second- and third-order interaction terms between sex, time, and dietary score.

[d]m1 is the full model, further adjusted for height, month of inclusion, physical activity, occupation, smoking status, educational level, monthly income, and cohabiting status, with second- and third-order interaction terms between sex, time, and dietary score.

FBDGs, food-based dietary guidelines; mPNNS-GS1, modified Programme National Nutrition Santé Guidelines Score; PNNS-GS2, Programme National Nutrition Santé Guidelines Score 2.

m1, associations were very similar, except that a positive association between mPNNS-GS1 and baseline BMI was observed. The same model was computed with PNNS-GS2 and mPNNS-GS1 in quintiles, and results are presented in S3 Table. In both models m0 and m1, we can see a linear pattern across both scores' quintiles, except for the intercept of mPNNS-GS1. This analysis was also replicated with continuous sPNNS-GS2 as the exposure, and results are presented in S4 Table. Coefficients were very similar to those of PNNS-GS2.

The results of the prospective association between dietary scores and the risk of overweight and obesity are presented in Table 5. After adjustment for confounding variables in both models m0 and m1, higher PNNS-GS2 was negatively associated with the risk of overweight and obesity, with a significant linear trend across quintiles whatever the model. Higher mPNNS-GS1 was negatively associated with the risk of overweight and obesity only in the fifth quintile or with dietary score considered as a continuous variable, but the association was weaker in model m1, especially for obesity, where it was no longer significant. However, caution is advised when interpreting the continuous HR for mPNNS-GS1, as the linearity hypothesis was not entirely verified for this score, as illustrated by the HRs across quintiles. Schoenfeld residuals graphical analysis and Grambsch–Therneau tests showed that the multivariable global assumption was not significantly violated. This analysis was performed again regarding the risk of morbid obesity (BMI $\geq 40$ kg/m$^2$) and is presented in S5 Table. Although a smaller number of events was observed, the results were even more discriminant: the association with the risk was significant for PNNS-GS2 in every quintile and in continuous variable analysis, but was not significant for any for mPNNS-GS1. The same analysis was also performed with sPNNS-GS2 as the outcome, and the results are presented in S6 Table. Hazard ratios were very similar to those of PNNS-GS2. Finally, this analysis was performed in the sensitivity subpopulation

**Table 5. Prospective association between PNNS-GS2 and mPNNS-GS1 and the risk of overweight and obesity—NutriNet-Santé study.** [a]

| Model | PNNS-GS2 | | | mPNNS-GS1 | | |
|---|---|---|---|---|---|---|
| | $N_{sane}/N_{case}$ | HR (95% CI) | *p*-Value[b] | $N_{sane}/N_{case}$ | HR (95% CI) | *p*-Value[b] |
| **Overweight: m0[c]** | | | <0.001 | | | 0.0002 |
| Q1 | 5,586/969 | 1 | | 6,158/802 | | |
| Q2 | 5,719/834 | **0.78 (0.71–0.86)** | <0.001 | 5,903/816 | 0.99 (0.90–1.09) | 0.87 |
| Q3 | 5,717/821 | **0.73 (0.66–0.80)** | <0.001 | 6,471/885 | 0.94 (0.85–1.03) | 0.20 |
| Q4 | 5,850/679 | **0.57 (0.52–0.63)** | <0.001 | 4,296/602 | 0.91 (0.82–1.01) | 0.09 |
| Q5 | 5,957/575 | **0.46 (0.41–0.51)** | <0.001 | 5,645/723 | **0.83 (0.75–0.93)** | <0.001 |
| 1 point[d] | 28,829/3,878 | **0.92 (0.91–0.93)** | <0.001 | 28,473/3,828 | **0.96 (0.94–0.98)** | <0.001 |
| 1 SD[d] | 28,829/3,878 | **0.75 (0.73–0.78)** | <0.001 | 28,473/3,828 | **0.94 (0.90–0.97)** | <0.001 |
| **Overweight: m1[e]** | | | <0.001 | | | 0.03 |
| Q1 | 5,586/969 | **1** | | 6,158/802 | | |
| Q2 | 5,719/834 | **0.80 (0.72–0.87)** | <0.001 | 5,903/816 | 1.02 (0.92–1.13) | 0.68 |
| Q3 | 5,717/821 | **0.75 (0.68–0.82)** | <0.001 | 6,471/885 | 0.99 (0.89–1.09) | 0.78 |
| Q4 | 5,850/679 | **0.59 (0.54–0.66)** | <0.001 | 4,296/602 | 0.97 (0.86–1.08) | 0.53 |
| Q5 | 5,957/575 | **0.48 (0.43–0.54)** | <0.001 | 5,645/723 | **0.90 (0.81–1.00)** | 0.05 |
| 1 point[d] | 28,829/3,878 | **0.92 (0.91–0.93)** | <0.001 | 28,473/3,828 | **0.98 (0.96–1.00)** | 0.04 |
| 1 SD[d] | 28,829/3,878 | **0.77 (0.74–0.79)** | <0.001 | 28,473/3,828 | **0.96 (0.93–1.00)** | 0.04 |
| **Obesity: m0[c]** | | | <0.001 | | | 0.02 |
| Q1 | 8,334/435 | 1 | | 8,698/358 | | |
| Q2 | 8,290/440 | 0.92 (0.80–1.05) | 0.22 | 8,472/367 | 0.98 (0.84–1.13) | 0.74 |
| Q3 | 8,390/368 | **0.71 (0.62–0.82)** | <0.001 | 9,534/446 | 0.98 (0.85–1.13) | 0.79 |
| Q4 | 8,400/316 | **0.57 (0.49–0.67)** | <0.001 | 6,412/284 | 0.90 (0.77–1.06) | 0.21 |
| Q5 | 8,461/253 | **0.43 (0.36–0.50)** | <0.001 | 8,227/341 | **0.84 (0.72–0.98)** | 0.03 |
| 1 point[d] | 41,875/1,812 | **0.91 (0.89–0.92)** | <0.001 | 41,343/1,796 | **0.96 (0.94–1.00)** | 0.02 |
| 1 SD[d] | 41,875/1,812 | **0.72 (0.68–0.76)** | <0.001 | 41,343/1,796 | **0.94 (0.90–0.99)** | 0.02 |
| **Obesity: m1[e]** | | | <0.001 | | | 0.80 |
| Q1 | 8,334/435 | 1 | | 8,698/358 | | |
| Q2 | 8,290/440 | 0.95 (0.83–1.09) | 0.45 | 8,472/367 | 1.04 (0.89–1.20) | 0.62 |
| Q3 | 8,390/368 | **0.75 (0.65–0.87)** | <0.001 | 9,534/446 | 1.09 (0.94–1.26) | 0.25 |
| Q4 | 8,400/316 | **0.62 (0.53–0.73)** | <0.001 | 6,412/284 | 1.02 (0.86–1.20) | 0.83 |
| Q5 | 8,461/253 | **0.47 (0.40–0.56)** | <0.001 | 8,227/341 | 0.98 (0.84–1.15) | 0.80 |
| 1 point[d] | 41,875/1,812 | **0.92 (0.90–0.93)** | <0.001 | 41,343/1,796 | 1.00 (0.97–1.03) | 0.97 |
| 1 SD[d] | 41,875/1,812 | **0.75 (0.71–0.79)** | <0.001 | 41,343/1,796 | 1.00 (0.95–1.05) | 0.97 |

[a]Bold values are significant.

[b]*p*-Values for whole models are computed using a linear trend test on quintile medians. *p*-Values for coefficients are computed using a Wald test for coefficient nullity.

[c]m0 is the base model, adjusted for sex, age, energy intake without alcohol, and number of completed 24-hour dietary records.

[d]The HR for 1 SD allows the comparison between the 2 scores, whereas the HR for 1 point gives an "absolute" estimation of the score effect. Yet, caution is advised when interpreting these values with the mPNNS-GS1 as the linearity hypothesis was not satisfyingly verified.

[e]m1 is the full model, further adjusted for height, month of inclusion, physical activity, occupation, smoking status, educational level, monthly income, and cohabiting status.

mPNNS-GS1, modified Programme National Nutrition Santé Guidelines Score; PNNS-GS2, Programme National Nutrition Santé Guidelines Score 2.

(with PNNS-GS2 and mPNNS-GS1), and the results are presented in S7 Table. Results were similar to the principal analysis, and associations were even stronger in some cases.

The one-model comparisons of scores for the multilevel models are presented in Table 6. Coefficients for PNNS-GS2 were significantly lower than for mPNNS-GS1 and sPNNS-GS2, which means that, compared to a high adherence to the 2001 guidelines and to the 2017

**Table 6. "Direct comparison" of the association of dietary scores with log(BMI)—NutriNet-Santé study.** [a]

| Model | Coefficient (95% CI) | p-Value[b] | Comparison p-Value[c] |
|---|---|---|---|
| **A** | | | |
| PNNS-GS2 | −0.056 (−0.058; −0.053) | <0.001 | <0.001 |
| mPNNS-GS1 | 0.030 (0.028; 0.032) | <0.001 | |
| PNNS-GS2 × time | −0.00066 (−0.00085; −0.00048) | <0.001 | 0.001 |
| mPNNS-GS1× time | −0.0013 (−0.00030; 0.000031) | 0.1 | |
| **B** | | | |
| PNNS-GS2 | −0.057 (−0.062; −0.052) | <0.001 | <0.001 |
| sPNNS-GS2 | 0.022 (0.017; 0.026) | <0.001 | |
| PNNS-GS2 × time | −0.00092 (−0.0013; −0.00051) | <0.001 | 0.007 |
| sPNNS-GS2 × time | 0.0019 (−0.00022; 0.00059) | 0.4 | |
| **C** | | | |
| sPNNS-GS2 | −0.049 (−0.051; −0.046) | <0.001 | <0.001 |
| mPNNS-GS1 | 0.028 (0.026; 0.030) | <0.001 | |
| sPNNS-GS2 × time | −0.00055 (−0.00074; −0.00036) | <0.001 | 0.03 |
| mPNNS-GS1× time | −0.0019 (−0.00036; −0.000020) | 0.03 | |

[a]Each model A, B, and C is a linear multilevel mixed model where pairs of dietary scores were standardized (by dividing by their SD) and considered as a continuous variable. This permits comparison of the effect of each dietary score while the other is fixed. The coefficient for the score represents its baseline effect, and interaction with time represents its slope effect. Models were adjusted for sex, age, energy intake without alcohol, number of completed 24-hour dietary records, height, month of inclusion, physical activity, occupation, smoking status, educational level, monthly income, and cohabiting status.

[b]p-Values were computed using a Wald test for coefficient nullity.

[c]p-Values were computed using a Wald test for coefficient equality between the 2 dietary score coefficients or the 2 interaction term coefficients.

mPNNS-GS1, modified Programme National Nutrition Santé Guidelines Score; PNNS-GS2, Programme National Nutrition Santé Guidelines Score 2.

**Table 7. "Direct comparison" of the predictive value of dietary scores for the risk of overweight and obesity—NutriNet-Santé study.** [a]

| Model | Overweight | | | Obesity | | |
|---|---|---|---|---|---|---|
| | HR | LRT[b] | Wald[c] | HR | LRT[b] | Wald[c] |
| **A** | | | <0.001 | | | <0.001 |
| PNNS-GS2 | 0.69 (0.66–0.72) | <0.001 | | 0.65 (0.61–0.69) | <0.001 | |
| mPNNS-GS1 | 1.19 (1.14–1.24) | <0.001 | | 1.27 (1.19–1.35) | <0.001 | |
| **B** | | | 0.001 | | | <0.001 |
| PNNS-GS2 | 0.74 (0.67–0.82) | <0.001 | | 0.55 (0.47–0.64) | <0.001 | |
| sPNNS-GS2 | 1.03 (0.93–1.13) | 0.6 | | 1.37 (1.18–1.58) | <0.001 | |
| **C** | | | <0.001 | | | <0.001 |
| sPNNS-GS2 | 0.71 (0.68–0.74) | <0.001 | | 0.71 (0.66–0.75) | <0.001 | |
| mPNNS-GS1 | 1.18 (1.13–1.24) | <0.001 | | 1.22 (1.15–1.31) | <0.001 | |

[a]In each model A, B, and C, pairs of dietary scores were standardized (by dividing by their SD) and considered as a continuous variable in the Cox proportional hazard models. This permits comparison of the effect of each dietary score on the outcome, while the other is fixed. Models were adjusted for sex, age, energy intake without alcohol, number of completed 24-hour dietary records, height, month of inclusion, physical activity, occupation, smoking status, educational level, monthly income, and cohabiting status.

[b]p-Values were computed using a LRT measuring the decrease in information when the variable is dropped from the adjusted model.

[c]p-Values were computed using a Wald test for coefficient equality between the 2 dietary scores.

LRT, likelihood ratio test; mPNNS-GS1, modified Programme National Nutrition Santé Guidelines Score; PNNS-GS2, Programme National Nutrition Santé Guidelines Score 2.

guideline principal recommendations only, a high adherence to the 2017 full guidelines is associated with a lower baseline BMI and a lower increase of BMI over time. The same conclusion could be reached when comparing sPNNS-GS2 to mPNNS-GS1.

For Cox models, one-model comparisons of scores are presented in Table 7. PNNS-GS2 was associated with a lower risk when the effect of other scores was fixed in all models. Conversely, for a fixed PNNS-GS2, mPNNS-GS1 was associated with a higher risk of both overweight and obesity, and sPNNS-GS2 was associated with a higher risk of obesity only. Comparison of sPNNS-GS2 with mPNNS-GS1 was similar to PNNS-GS2 versus mPNNS-GS1, but with a slightly lower strength of association. PNNS-GS2 was also compared to AHEI-2010 for risk of overweight ($HR_{PNNS-GS2}$ = 0.83 [95% CI 0.79–0.87], $HR_{AHEI-2010}$ = 0.89 [95% CI 0.85–0.94]) and obesity ($HR_{PNNS-GS2}$ = 0.82 [95% CI 0.76–0.88], $HR_{AHEI\ 2010}$ = 0.88 [95% CI 0.82–0.94]), with the Wald test $p$-value being 0.06 for overweight and 0.16 for obesity (S8 Table). Likewise, this comparison was performed in a multilevel model, and PNNS-GS2 was significantly associated with a lower baseline BMI than AHEI-2010 ($p < 0.001$) but with a higher BMI increase over time ($p < 0.001$).

## Discussion

In the present study, the adherence to the 2017 French FBDGs assessed by PNNS-GS2 was associated with a significantly lower risk of developing overweight (up to 50% in Q5 versus Q1) and obesity (up to 60% in Q5 versus Q1). These findings were robust as PNNS-GS2 was associated with a lower baseline BMI but also with a lower increase of BMI over time. Comparison of the magnitude of associations between dietary scores and BMI evolution and overweight and obesity risk showed a clear superiority of the 2017 FBDGs, both with all recommendations (PNNS-GS2) and with only principal ones (sPNNS-GS2), over the 2001 FBDGs (mPNNS-GS1). The association of PNNS-GS2 with weight outcomes was also significantly stronger than that of its simplified version, sPNNS-GS2, and comparable to that of AHEI-2010.

Concerning the risk of overweight and obesity, the association with mPNNS-GS1 appeared to be marginal and only noticeable for participants with the highest level of adherence (Q5), which could mean that following previous FBDGs would mostly be beneficial if highly followed. Association with PNNS-GS2 was stronger than with mPNNS-GS1 and with a linear trend across quintiles, illustrating that a higher adherence to the 2017 FBDGs could be beneficial regardless of level compared to the lowest adherence.

In the one-model comparisons of scores, the adjustment for PNNS-GS2 reversed the direction of the association with risk of overweight and obesity for mPNNS-GS1 from a rather negative to a rather positive association. Thus, we can consider as particularly "healthy" both the removal of some mPNNS-GS1 components (for instance promotion of "all meat, fish, and eggs" and cereals without specificity) and the introduction of certain components into PNNS-GS2 (promotion of nuts and legumes and discouraging of red and processed meat). Hence, having a high mPNNS-GS1 appears to be deleterious when PNNS-GS2 is fixed.

The same phenomenon was observed for sPNNS-GS2 when adjusting for PNNS-GS2, but only for obesity. Hence, it is likely that secondary recommendations are of particular importance for obesity prevention. As secondary recommendations pertained mainly to organic food consumption, this is in accordance with previous literature, which found out that chemical exposure was linked with obesity and type II diabetes [11], and with a previous prospective study conducted in the same cohort [10].

On the other hand, comparison of PNNS-GS2 with AHEI-2010 does not show such a qualitative difference. Indeed, since both dietary scores remained protective for overweight and

obesity after adjustment for the other in Cox models, we can assume that they may contain complementary components. Multilevel models provided similar findings, as the comparison of baseline association and association over time were inverse. Further investigation would be required to identify the complementarities, but this finding is still interesting considering future improvements of national FBDGs.

Beyond promotion of organic food consumption, there are several differences in food group consumption that could explain the difference between mPNNS-GS1 and PNNS-GS2 for the risk of overweight and obesity. The most important one is likely to be red and processed meats, which are specifically discouraged in the 2017 FBDGs but which were not included in 2001, and which are now acknowledged to be associated with obesity and diabetes incidence [12,43–45]. Nuts and legumes, recommended only in the 2017 FBDGs, have also been associated with a reduction of weight gain and obesity/overweight risk [46,47] and are important components of the Mediterranean diet, which is known to be protective for abdominal obesity [48,49]. As well, promotion of whole-grain foods in the 2017 FBDGs is more noticeable than in the 2001 guidelines, and their intake has been inversely associated with weight gain [50] and positively associated with a lower risk of overweight and obesity [49]. Artificially sweetened beverages are also specifically addressed in the 2017 FBDGs and have been associated with a higher risk of excessive weight gain and metabolic syndrome [51], although a recent meta-analysis underlined methodological shortcomings in studies addressing non-sugar sweeteners [52].

Our results can be compared to previous studies conducted in another French sample, SU. VI.MAX. Indeed, the present estimation of the association between mPNNS-GS1 and overweight and obesity incidence was very similar to that found by Kesse-Guyot et al. [53]. We also observed similar findings about the role of PNNS-GS1 in weight change and obesity risk as reported in men by Lassale et al. [54] and Assmann et al. [55], suggesting external validity of our findings.

Some limitations to our study should still be highlighted. Indeed, selection bias could have occurred as these analyses were drawn from the NutriNet-Santé cohort, whose participants may be more interested in nutrition than the general population, limiting external validity. This bias may have been strengthened by the further selection of our sample within the cohort. Such bias might have excluded participants with poor diet, so our estimations could be underestimated, with our lowest quintile being healthier than the lowest quintile of the whole population. Also, our data were self-declared, although they were validated against clinical examination [32] and biomarkers [28]. In the Cox models, BMI had to be categorized using cutoffs, which are rather arbitrary, although we used the official WHO cutoffs that are commonly used and therefore have good external validity and are widely understood. Both of the above limitations may have led to decrease in statistical power. Besides, even though the study was designed as prospective, reverse causality may not be entirely ruled out, as obesity is a complex process. The follow-up could also be somehow questioned, as people with the longest follow-up may have been particularly compliant and health conscious, with therefore loss to follow-up potentially associated with the outcome, which might lead to underestimation in our results. In the one-model comparisons of scores, multicollinearity could be considered an issue. However, since it only affects type II errors (false negative associations), its effect is very low in this study of rather high statistical power. BMI was used to evaluate overweight and obesity, which is highly subject to misclassification depending on age, sex, and fat repartition [56,57]. Better estimation tools have been recently proposed, such as relative fat mass, which allows for better prediction of adiposity, but could not be used in our study as it requires clinical waist circumference measurement [58]. Residual confounding may still have affected the strength of the association. Indeed, some factors, such as ethnicity, which influences body

weight status and diet quality [59], were not taken into account. Lastly, we considered the effect of early (first 2 years) diet quality in a strictly prospective design, but we cannot assume that some people have not changed their eating habits.

Despite these limitations, our study documented strong inverse associations between PNNS-GS2 and the risk of overweight and obesity, and provided consistent evidence about the superiority of the 2017 FBDGs over the 2001 guidelines in terms of prevention. An important strength of this work is its prospective design and its acceptable follow-up (median of 6 years). Our dietary data were also highly accurate, with an average number of 24-hour records of 8 per individual, thus accounting for daily variation. Moreover, the large size of the sample provided reliable statistical power. Finally, PNNS-GS2 is one of the few dietary scores that include recent concepts such as reducing exposure to diet-related contaminants and reducing consumption of animal products.

In conclusion, our findings suggest that following the 2017 FBDGs tends to be associated with a lower risk of overweight and obesity. The magnitude of the associations and the results of the "direct comparison" reinforce the validity of the updated recommendations. Since overweight and obesity are major risk factors of major chronic diseases, adherence to the new French FBDGs may help to prevent chronic diseases, and further investigations will be carried out to test this hypothesis.

## Supporting information

**S1 STROBE checklist.**
(DOCX)

**S1 Table. Characteristics of the participants by quintile of mPNNS-GS1—NutriNet-Santé study, $N$ = 54,089.**
(DOCX)

**S2 Table. Comparison of baseline characteristics of the participants in the whole NutriNet-Santé study ($N$ = 138,014) and in our selected population ($N$ = 54,089).**
(DOCX)

**S3 Table. Longitudinal evolution of log(BMI) by quintile of PNNS-GS2 and mPNNS-GS1 —NutriNet-Santé study.**
(DOCX)

**S4 Table. Longitudinal evolution of log(BMI) as a function of sPNNS-GS2—NutriNet-Santé study.**
(DOCX)

**S5 Table. Prospective association between PNNS-GS2 and mPNNS-GS1 and the risk of morbid obesity (BMI > 40 kg/m$^2$)—NutriNet-Santé study.**
(DOCX)

**S6 Table. Prospective association between sPNNS-GS2 and the risk of overweight and obesity—NutriNet-Santé study.**
(DOCX)

**S7 Table. Prospective association between PNNS-GS2 and the risk of overweight and obesity in a subpopulation for sensitivity analysis—NutriNet-Santé study.**
(DOCX)

**S8 Table. "Direct comparison" of the predictive value of PNNS-GS2 and AHEI-2010 for the risk of overweight and obesity—NutriNet-Santé study.**
(DOCX)

## Acknowledgments

The authors warmly thank all the volunteers of the NutriNet-Santé cohort. We also thank Younes Esseddik, Frédéric Coffinieres, Thi Hong Van Duong, Paul Flanzy, Régis Gatibelza, Jagatjit Mohinder, and Maithyly Sivapalan (computer scientists); Cédric Agaesse (dietitian); Julien Allègre, Nathalie Arnault, Laurent Bourhis, Véronique Gourlet, PhD, and Fabien Szabo de Edelenyi, PhD (data-managers/biostatisticians); Roland Andrianasolo, MD, and Fatoumata Diallo, MD (physicians); and Nathalie Druesne-Pecollo, PhD (operational coordinator), for their technical contribution to the NutriNet-Santé study.

## Author Contributions

**Conceptualization:** Dan Chaltiel, Emmanuelle Kesse-Guyot.

**Data curation:** Dan Chaltiel.

**Formal analysis:** Dan Chaltiel.

**Funding acquisition:** Serge Hercberg.

**Investigation:** Chantal Julia, Mathilde Touvier, Serge Hercberg, Emmanuelle Kesse-Guyot.

**Methodology:** Dan Chaltiel, Emmanuelle Kesse-Guyot.

**Software:** Dan Chaltiel.

**Supervision:** Emmanuelle Kesse-Guyot.

**Validation:** Dan Chaltiel, Chantal Julia, Moufidath Adjibade, Mathilde Touvier, Serge Hercberg, Emmanuelle Kesse-Guyot.

**Writing – original draft:** Dan Chaltiel, Emmanuelle Kesse-Guyot.

**Writing – review & editing:** Dan Chaltiel, Chantal Julia, Moufidath Adjibade, Mathilde Touvier, Serge Hercberg, Emmanuelle Kesse-Guyot.

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
