## [Decision Letter · Decision Letter 0]

10 Sep 2019

Dear Dr. Chaltiel,

Thank you very much for submitting your manuscript "Adherence to the French dietary guidelines and risk of overweight and obesity" (PMEDICINE-D-19-02728) for consideration at PLOS Medicine. 

[LINK]

In light of these reviews, I am afraid that we will not be able to accept the manuscript for publication in the journal in its current form, but we would like to consider a revised version that addresses the reviewers' and editors' comments. Obviously we cannot make any decision about publication until we have seen the revised manuscript and your response, and we plan to seek re-review by one or more of the reviewers. 

We expect to receive your revised manuscript by Oct 01 2019 11:59PM. Please email us (plosmedicine@plos.org) if you have any questions or concerns.

We look forward to receiving your revised manuscript. 

Sincerely,

Adya Misra, 

Senior Editor 

PLOS Medicine

plosmedicine.org

We note some serious issues raised by the referees ad urge you to address these in full. Below are some specific editorial points and queries:

Title – this needs to be re-worked and to include a study descriptor. I suggest:

Association of adherence to the 2017 French dietary guidelines and risk of overweight and obesity; a prospective cohort study.

Abstract – please state which cities participants are from and please also add some summary information on articipant characteristics; please ensure in the abstract as well as throughout and also in Tables that p values are used where data is quantiified with 95%Cis.

Abstract – please restructure with 3 sections: Background, Methods and Findings and Conclusions and provide a sentence on the limitations of the study as the final sentence of the Methods and Findings section.

References in the main text – please use square brackets and remove superscript

Please ensure all questionnaires are available or submitted as Sup Files.

Did your study have a prospective protocol or analysis plan? Please state this (either way) early in the Methods section.

c) In either case, changes in the analysis—including those made in response to peer review comments—should be identified as such in the Methods section of the paper, with rationale.

Please provide a STROBE reporting guidelines (submitted as a Supp file and ensure paragraph and sections are used instead of page numbers – these will change in the event of publication.)

Comments from the reviewers:

Reviewer #1: I confine my remarks to statistical aspects of this paper. I think there is a very good paper here, but, to get to that paper requires some fairly substantial changes.

The biggest change is that the supplemental analysis (using multilevel models) should be the main analysis and what is now the main analysis should either be dropped or be a supplemental analysis. There are several reasons for this:

1. BMI is a terrible measure of obesity as it doesn't account for things such as muscle mass, body proportion and so on. Categorizing BMI into "normal", "overweight" and "obese" makes it even worse, as it increases both type I and type II error, decreases the precision of estimates and introduces a sort of "magical thinking" - that something amazing happens at BMI = 25 and that 24.9 is both different from 25.1 and similar to (say) 20, while 25.1 is similar to 29.9, which is different from 30.1, while 30.1 is similar to the maximum value.

2. While the above is true, change in BMI is a good measure of change in obesity, since body proportion is constant and muscularity is unlikely to change much in the course of a study.

3. MLMs can deal with missing data much better than Cox models, using existing data even on people with missing data.

In fact, since the authors have already done both analyses, they could provide a service by showing how the MLM were better with these data (they already do this, but it isn't highlighted).

And, yes, I know BMI is used a lot and so are the categorized versions. They shouldn't be. This paper (redone as above) could be evidence of why not.

Other issues:

Using age instead of time on study as the time scale is appropriate. However, this is not widely known and should be highlighted -that is, say why you did this and cite some papers. 

Line 67-68: This sentence makes no sense. You can't "randomly attribute" days to weekdays and weekends. I'm not sure what is meant here. Perhaps that "weekday" vs. "weekend" was taken into account somehow? 

Line 82-84: I don't understand this sentence. Do you mean that it was dummy coded?

Line 130-131 There is no reason to categorize an independent variable. In *Regression Modelling Strategies* Frank Harrell gives 11 reasons why this is a bad idea and summarizes "nothing could be more disastrous". i gave a graphical demonstration of this in a blog post: https://medium.com/@peterflom/what-happens-when-we-categorize-an-independent-variable-in-regression-77d4c5862b6c

Line 136-138: I would abandon the categorized version and present more detailed results on the continuous one, especially as you used a spline (good idea! But results of spline models need interpretatlon). I saw no discussion of this.

Line 144-146: I don't understand. You assigned the median of each quintile and then treated it as continuous? Why? You can use the score itself.

Line 149-150: Please show how each variable was operationalized. This is now only shown in table 2. And income and activity should not be categorized (see above) (This seems to have been done for both the MLM and the Cox models)

Line 177 - Was it really SAS 7.15? They are on version 9.4 now. 7.15 is many years out of date.

Figure 1 - The sample that was used is a small proportion of the total sample. Excluding pregnant women is fine, but al the other exclusions are troublesome - Why did so many people have missing data? How were the people with missing data diferent from those with complete data? In particular, I think under-reporters and people without follow up BMI are likely to be different in important ways from those with accurate reporting and follow up BMI (I'd bet people who gained a lot of weight would be much more likely to skip the follow up BMI). Also, what are the boxes with 3907 and 5174 that come after the "all necessary data" box?

The authors do mention bias, but this needs more than one sentence.

Table 4 and supp tables: For the continuous model, if you used a spline then the 1 point and 1 SD numbers aren't clear. The effect of 1 point or 1 SD will be different at different levels of the IV. 

Sorry to be so negative - but I think that, if these issues are addressed, the paper could be valuable both substantively and methodologically. 

Reviewer #2: The current manuscript reports a new dietary profile score which has been derived to assess the impact of adherence to new dietary guidelines in France. The novel aspect is the development of a scoring system based on a specific update to a dietary policy. The new scoring system was compared to a previous tool with and without physical activity. The table 1 compares the PNNS-GS scoring system with the previous version, it would appear that some of the additional lifestyle scoring is not based on nutritional change associated with NDC reduction but on wide food policy. For example, the consumption of organic produce. I did not think there is any reported health benefit associated with the consumption of organic products. However, I do understand that the scoring is related to the policy that was derived by the High Council for Public Health in France and the score is derived to reflect this. Also, I wonder if this has produced a bias in the distribution of the sample- the quintile table suggest that those in Q5 older, higher educational achievement, greater percentage retired and have a higher income.

I do not understand why there is such a sex bias with over three times as many women as men? This must make the data diffiult to intepret as a cohort as a whole as it is dominated by one sex.

A strength of the study is size with over 53,000 who report over 7 dietary records with a 6 year follow up. However, the relationship described between body weight and the PNNs-GS2 are not novel many other have demonstrated compliance to dietary guidelines is associated with lower body weight e.g. Wang etal BMJ 2018. This in my main criticism am unclear other than the guideline based score what is novel about this piece of work.

Reviewer #3: This study aimed to assess the association between adherence to French dietary guidelines, past and present, with development of overweight and obesity in a French internet-based cohort.

The manuscript is generally well-written, however I urge the authors to consult the STROBE guidelines and structure their discussion section accordingly. Also, I suggest checking their reference management system, and setting the language to English, rather than French.

Major concerns:

The most noticiable differences between the 2001 dietary guidelines score and the new dietary guidelines score are the differences in score weighting of adherence to the components (eg salt is rewarded differently in the two scores), and that organic food choices are prominent in the subcomponents of the new score. My main concern is that differences in socioeconomic position related to the ability to purchase organic food is built into the new score, and that this underlying residual confounding (or even a form of reverse causality) cannot be adjusted away in the statistical models. Although the authors mention having calculated a slimmed down version of the new score, I was unable to find results assessing the association between the slim score and the chosen outcomes. This is important information for readers, and the public.

The introduction dwells on obesity as a public health problem, with which I agree. However, some statements are very strong, in particular on lines 13-15, that overweight can be considered a marker of health (also, the reference to a paper on dietary patterns is an unusual choice). This ignores a large body of literature showing that the lowest risk of mortality is in the low overweight range, a body of literature with which many readers will be familiar. Please rebuild this argument using the literature more critically.

Similarly, the body of literature describing studies of dietary guidelines in relation to health is wider than the French and US studies cited (refs 14-16). For example Dutch and Danish guidelines, to just mention two other national FBDG that have been evaluated in relation to health. Please consider and rebuild this argument more critcally, also. Is a discussion of validation of the score relevant in the introduction to this paper?

Methods:

Sample selection.

Do the 115536 participants shown in the first box of the flow chart constitute the entire NNS cohort, or were exclusions performed to reach this number. This is important for the discussion of selection bias, as well as external validity.

Exposure:

How the 24hr DRs were converted to the exposure was unclear to me. More specifically, did participants contribute with varying numbers of dietary recalls, or just the first one? (What is the implication of this for episodically eaten foods?)

Did the authors consider a DAG when building their model? A figure illustrating their assumptions about the interrelationships of the covariates would be useful.

Statistics:

Please clarify when participants' baseline started, and whether any censoring took place. How were the varying numbers of outcome questionnaires (and varying time between them) taken into account? Would a regression model for interval censored data be more appropriate than a Cox model?

Were models adjusted for alcohol intake?

Please also clarify how tests for interaction were conducted.

Discussion:

At a first reading, it appears that the results of the mixed model somewhat contradict the conclusions based on the Cox models. Please also put the results for the 2001 score into context.

In the limitations section, please consider how the various possible biases that are listed would affect the results - might they be over or understimated? Please also dwell a little more on who these results might be generalizable to. The discussion section mentions selection bias and generalizability in one sentence, but I would argue that these are two different concepts. The first is about whether the results represent the cohort. The second is about who the cohort represents. With so may excluded particpants, it could be argued that these results apply mainly to people who are good at regularly filling out questionnaires about sensitive subjects such as weight. It is important that the authors either acknowledge or counter this concern.

Minor concerns:

The acronyms for these dietary scores are long and quite similar. Please consider enhancing the reader's experience by chosing more intuitive acronyms.

Abstract, and elsewhere:

Terms such as "reduction in risk" and "decreased risk", as well as "increased risk" are used several places in the manuscript. Because it appears that the authors have evaluated baseline diet at a single timepoint, they are not evaluating changes in exposure and subsequent changes in risk. It is therefore more appropriate the discuss "higher" and "lower" risks.

The introduction, paragraph 2, mixes causes of obesity with registering obesity as a disease (ie obesity as an outcome). Please consider the argument of this paragraph and restructure accordingly.

The AHEI 2010 does not measure adherence to the US dietary guidelines, but to a Harvard School of Public Health-constructed healthy diet pattern. The HEI, for example the HEI 2010, measures adherence to the US guidelines.

The multilevel model is very interesting, but the modelling choices are unclear to me, and the language describing the methods uses different terms (base and full model, for example) than the previous section.

[LINK]

---

## [Decision Letter · Decision Letter 1]

14 Nov 2019

Dear Dr. Chaltiel,

Thank you very much for re-submitting your manuscript "Association of adherence to the 2017 French dietary guidelines and weight gain; a prospective cohort study" (PMEDICINE-D-19-02728R1) for review by PLOS Medicine.

I have discussed the paper with my colleagues and the academic editor and it was also seen again by xxx reviewers. I am pleased to say that provided the remaining editorial and production issues are dealt with we are planning to accept the paper for publication in the journal.

[LINK]

We look forward to receiving the revised manuscript by Nov 21 2019 11:59PM. 

Sincerely,

Adya Misra, PhD

Senior Editor 

PLOS Medicine

plosmedicine.org

Requests from Editors:

Please add a few words to your data statement to explain the criteria by which applications for data will be assessed, e.g., for compliance with the study's ethics approval. 

Please modify your title to match journal style. We suggest: "Adherence to the 2017 French dietary guidelines and adult weight gain: a cohort study". 

Please refer to obesity as a condition rather than a disease throughout to avoid use of potentially stigmatising labels 

We suspect that you are reporting a retrospective analysis of prospectively-gathered data. Therefore, please replace the word "prospective" early in your abstract, and at any other points in the paper. 

Regarding the summary of limitations in your abstract, we suggest adapting the wording to "Study limitations include possible selection bias, reliance on participant self-report, use of arbitrary cutoffs in data analyses, and residual confounding." or similar. 

In your author summary, please convert "key-lever" to "lever". 

Please soften the wording of the third-last summary point to "... can be expected to improve ..." or similar. 

Please remove "important" from the penultimate line of your abstract. 

At line 5, please make that "estimated to be". 

Line 12- should start with “obesity has already been proven…” and line 14 should contain “..and is associated with quality of life ...”

Line 19- this sentence reads awkwardly, “human genetics has not changed significantly in the past decades” and we suggest it is removed

Please ensure it is consistently PNNS-GS2 throughout the text

Around line 63, please state whether any non-prespecified analyses were performed, and highlight these where they are presented in the text. 

Line 68 should read “…was recorded by...”

Line 203 contains a reference to “data not shown” which is not permitted as per PLOS Data policy. Please provide the data in the results section or as supplementary information or remove this sentence. 

Line 253 contains “data not tabulated” – please provide these data in a table format either in the main text or SI files 

Please correct the ?reference call out at line 80. 

At line 257, please make that "Discussion". 

Please replace "p<0.0001" with "p<0.001" or an exact value, throughout. 

Please adapt formatting of reference call-outs throughout your text, e.g., "... mortality [1,4]."

Please remove trade marks from your paper. 

Unsure if this statement in the discussion adds anything of relevance to the study? “the more someone is gaining weight, the more he/she tends to diet to lose the gained weight to conform to social norms of body weight” as such we would suggest removing this

Please clarify how a highly selected population and participants lost to follow-up can cause an underestimation of the effect of diet adherence as opposed to overestimation? 

Line 337-339 please rephrase this sentence for grammar and usage. Please consider rephrasing to “…such as reduced exposure to diet-related contaminants and fewer animal products” 

Please remove page numbers from STROBE checklist as these are likely to change during publication 

Comments from Reviewers:

Reviewer #1: The authors have addressed my c0ncerns and I now recommend publication. 

Reviewer #3: The authors have taken my previous comments into account, and in my opinion the manuscript is much clearer in argumentation and documentation now. I do, nevertheless, have some minor comments on the current version.

Introduction: The estimate of the cost of obesity - please clarify whether this refers to France alone? 

Discussion: 

line 271: I think it is a little misleading to claim that adherence is beneficial regardless of baseline levels, because it implies that baseline changes (ie adherence changes). I suggest something like "...regardless of level compared to the lowest adherence."

line 316: I don't understand how using WHO's cutoffs lowers statistical error. I do, however, understand the wish to present results using cutoffs with which readers may be familiar, despite the inbuilt statistical errors. I suggest rewriting this sentence to reflect this.

[LINK]

---

## [Editor Report · Decision Letter 2]

2 Dec 2019

Dear Dr Chaltiel, 

On behalf of my colleagues and the academic editor, Dr. Christina Catherine Dahm, I am delighted to inform you that your manuscript entitled "Adherence to the 2017 French dietary guidelines and adult weight gain: a cohort study" (PMEDICINE-D-19-02728R2) has been accepted for publication in PLOS Medicine. 

PRODUCTION PROCESS

PRESS

PROFILE INFORMATION

Thank you again for submitting the manuscript to PLOS Medicine. We look forward to publishing it. 

Best wishes, 

Adya Misra, PhD

Senior Editor 

PLOS Medicine

plosmedicine.org